# Generation of axenic *Aedes aegypti* demonstrate live bacteria are not required for mosquito development

Maria A. Correa[1], Brian Matusovsky[2], Doug E. Brackney[1] & Blaire Steven[3]

The mosquito gut microbiome plays an important role in mosquito development and fitness, providing a promising avenue for novel mosquito control strategies. Here we present a method for rearing axenic (bacteria free) *Aedes aegypti* mosquitoes, consisting of feeding sterilized larvae on agar plugs containing a high concentration of liver and yeast extract. This approach allows for the complete development to adulthood while maintaining sterility; however, axenic mosquito's exhibit delayed development time and stunted growth in comparison to their bacterially colonized cohorts. These data challenge the notion that live microorganisms are required for mosquito development, and suggest that the microbiota's main role is nutritional. Furthermore, we colonize axenic mosquitoes with simplified microbial communities ranging from a single bacterial species to a three-member community, demonstrating the ability to control the composition of the microbiota. This axenic system will allow the systematic manipulation of the mosquito microbiome for a deeper understanding of microbiota-host interactions.

[1] Center for Vector Biology and Zoonotic Diseases, The Connecticut Agricultural Experiment Station, New Haven 06511 CT, USA. [2] Department of Ecology and Evolutionary Biology, Yale University, New Haven 06511 CT, USA. [3] Department of Environmental Sciences, The Connecticut Agricultural Experiment Station, New Haven 06511 CT, USA. Correspondence and requests for materials should be addressed to D.E.B. (email: doug.brackney@ct.gov) or to B.S. (email: blaire.steven@ct.gov)

t is increasingly clear that most, if not all, multicellular organisms live in association with a complex assemblage of microorganisms (i.e., microbiome) composed of bacteria, viruses, fungi, and archaea. While these communities can be found in every habitable organ, for most complex organisms the vast majority of microbes reside in the digestive tract. Because of the biomass and complexity of the indigenous gut microbiome, as well as its close association with the host, it is frequently considered an additional major organ[1,2]. Consequently, the influence of microbiota on host biology has garnered considerable attention[3,4]. These studies have revealed a link between the microbiome and a wide array of physiological states in mammals, including obesity[5], diabetes[6], and behavior[7,8]. Furthermore, the microbiome has been implicated in playing a significant role in the development and function of the immune system and auto-immune disorders[9,10].

Invertebrates also harbor a diverse microbiome[11,12] that has been linked to a number of phenotypic outcomes, such as host-mating preference[13] and embryonic development[14]. It is clear from these studies that the microbiome can have profound effects on host physiology and health. Mosquitoes are important disease vectors for a number of human pathogens, including arboviruses, protozoa, and nematodes, that pose a significant public health threat. Due to the lack of an effective vaccine for many of these pathogens and an increase in insecticide resistance in mosquitoes, the development and implementation of novel mosquito control strategies will be necessary to curtail their public health impact. The mosquito microbiome is emerging as a potential tool in this effort[15]. A number of descriptive studies, primarily focused on the bacterial components of the microbiome, have determined that it is relatively simple, typically composed of 10–70 bacterial strains, the majority of which are members of the phylum Proteobacteria, specifically the family *Enterobacteriaceae*[16,17]. Based on the similarity in composition between the microbiome of mosquito larvae and the water they inhabit, it has been proposed that mosquitoes largely acquire their gut microbiota from the aquatic environment[18]. Further evidence suggests that at least some of the larval microbiome is transstadially transmitted to the adult after pupation[19]. While these studies demonstrate that environmental microorganisms readily colonize mosquitoes and these associations can be stable over the entire lifespan, the role these microbes play in mosquito development and biology is less clear.

Most mosquito-borne pathogens must infect or pass through the mosquito midgut prior to being transmitted. Not surprisingly, the complex interplay between pathogens, the mosquito midgut, and its associated microbiome have garnered considerable attention. For instance, it is known that bacterial load and/or microbiome community composition can significantly affect *Anopheles spp.* mosquito susceptibility to *Plasmodium* infection[20,21], and *Aedes aegypti* susceptibility to dengue virus is influenced by the gut microflora[22,23]. Microbiota have differing effects on vector competence in mosquitoes, with particular isolates either positively or negatively influencing mosquito infection rates depending on the species or bacterial strain[22,24–26]. Taken together, these observations demonstrate that the composition and structure of the microbiome can affect the ability of mosquitoes to acquire and transmit disease. Yet, because no current method exists to systematically manipulate the microbiome, these studies are by definition correlational in nature. Furthermore, it is difficult to determine the impact of the microbiota on mosquito-pathogen interactions because many of these studies have relied upon antibiotic clearance of the bacterial communities. Recent reports show that the mosquito microbiome often contains antibiotic resistant bacteria[27], antibiotics do not fully clear the gut microbiota, but rather cause a dysbiosis[28], and extended use of antibiotics can cause toxicity and mitochondrial dysfunction[29].

Consequently, these models cannot be truly considered bacteria-free and do not address possible interactive effects between bacterial reduction and antibiotic exposure on mosquito biology and vector competence.

Systematic manipulation of the mosquito microbiome would be greatly facilitated by the existence of a microbiome-free, or axenic, mosquito[30]. Furthermore, the development of an axenic model could act as a blank template on which a microbiome of known composition could be imprinted, also known as a gnotobiotic organism[31,32]. Axenic rearing techniques have already been developed for a number of model organisms, including *Drosophila melanogaster*, *Caenorhabditis elegans*, and mice[30]. Early attempts to rear axenic mosquitoes reportedly obtained adults free from bacteria using a growth media of essential vitamins and nutrients[33]. However, these studies lacked the modern molecular-based techniques that can detect microorganisms that are recalcitrant to laboratory cultivation; as is frequently cited, the majority of microorganisms in nature are unculturable[34]. In this regard, there is some uncertainty as to whether these mosquitoes were truly axenic. In fact, a series of recent studies have reported that mosquitoes require a live bacterial symbiont for development[27,35,36]. Yet, studies describing the necessity of live bacteria generally ignored the role of microflora in supplying essential nutrients to the host. Thus, there is somewhat of a contradiction in the literature; either mosquitoes are unique from *Drosophila*, *C. elegans*, and mice in requiring a live bacterial symbiont for development, or nutritional conditions sufficient to rear axenic mosquitoes have yet to be documented.

In this study, we test a variety of reported methods and develop novel practices for the rearing of *Aedes aegypti* mosquitoes free of living bacteria. By maintaining larvae hatched from surface sterilized eggs on a diet consisting of agar plugs containing liver and yeast extract, with or without heat-inactivated bacteria, we demonstrate a means to rear axenic mosquitoes to adulthood and into the next generation. Biometric comparisons reveal that developmental times differ between axenic mosquitoes and gnotobiotic mosquitoes colonized by *Escherichia coli* or an *Ae. aegypti* colony-derived microbiome. Furthermore, we demonstrate the utility of this system by imprinting larvae and adult mosquitoes with simplified microbiomes ranging from a single species to a simplified three-member community. Our data represents a methodological advancement in the field of mosquito microbiome research and provides a much-needed tool to elucidate the role of microbiota in mosquito physiology and pathogen susceptibility. Furthermore, our results challenge our current understanding of the interaction between the microbiome and mosquito development.

## Results

**Testing mosquito diets to support axenic mosquito growth.** In order to define the nutritional requirements needed to support mosquito development, we tested multiple mosquito diets (Table 1; lines 1–4). Mosquito diets free of living bacteria generally resulted in widespread mortality or stalled larval development. In contrast, the majority of gnotobiotic larvae (colonized by *E. coli* strain K12) readily pupated and emerged as adults. Yet, supplementing the mosquito diet with heat-killed or sonicated *E. coli* cells in the medium either failed to rescue development or resulted in larval mortality (Table 1; lines 5–8). Similar results were obtained when the medium was supplemented with yeast extract (Table 1; lines 9,10). A mixture of amino acids and vitamins commonly used for cell culture was also assessed for its ability to support larval growth. Larvae failed to develop at low concentrations of the added nutrients, whereas the mixture was lethal to the larvae at high concentrations (Table 1, lines 11, 12).

**Table 1 Effect of sterile diets on larval development**

| Diet | Axenic | Gnotobiotic[a] |
|---|---|---|
| *Base media* | | |
| 1. Liver:yeast extract (LY) | Stalled[b] | Pupae[c] |
| 2. Luria broth | Stalled | Dead |
| 3. Larval growth medium | Dead | Pupae |
| 4. Fish food | Stalled | Pupae |
| *Supplemented media (LY base)* | | |
| 5. 1 ml *E. coli* sonicate | Stalled | NA[d] |
| 6. 5 ml *E. coli* sonicate | Dead | NA |
| 7. 1 ml *E. coli* autoclaved | Stalled | NA |
| 8. 5 ml *E. coli* autoclaved | Dead | NA |
| 9. 1 ml 1% yeast extract | Stalled | Pupae |
| 10. 1 ml 10% yeast extract | Dead | Dead |
| 11. Amino acid vitamin mix (0.5 ×) | Stalled | NA |
| 12. Amino acid vitamin mix (1 ×) | Dead | NA |
| 13. Active yeast culture | Pupae | Pupae |
| *Agar plugs* | | |
| 14. Autoclaved *E. coli* and LY base | Pupae | NA |
| 15. Autoclaved *E. coli* no LY base | Stalled | NA |
| 16. LY base no *E. coli* | Pupae | NA |
| 17. LY base (2 ×) | Dead | NA |

[a]Gnotobiotic larvae were colonized by *E. coli* by adding a 10 μl aliquot of an overnight culture to the larval media
[b]Larva did not develop beyond the initial L1 stage
[c]Indicates full development of larvae to pupae
[d]Not tested

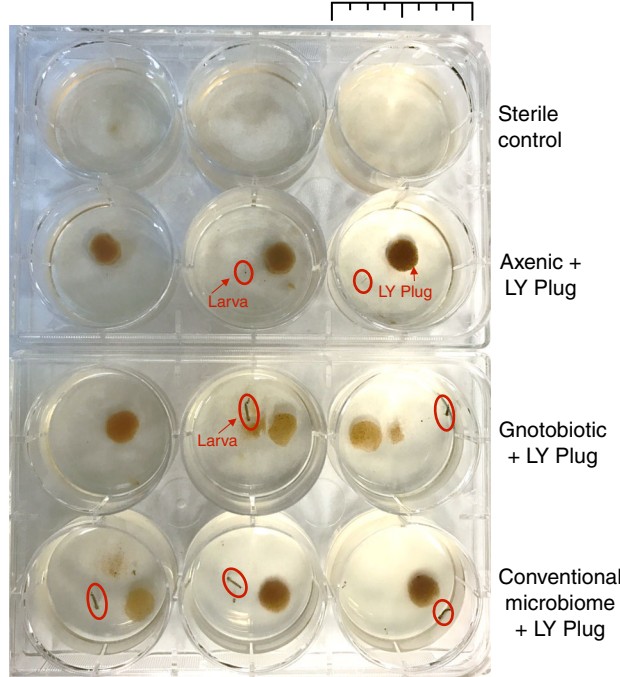

**Fig. 1** Experimental setup. Individual larvae were reared in the wells of a six-well plate. This picture was taken 5 days post hatching. The sterile control group consisted of autoclaved liver:yeast extract not embedded in an agar plug, which does not support larval development (individual larvae in this group are too small to see at this scale). Any development of sterile control larvae was taken as a sign of contamination and the experiment was discarded. Because of the delay in development, the axenic larvae are smaller than the gnotobiotic or CM groups. The scale bar at the top of the figure denotes 3.5 cm. The red arrow and text indicate the LY plug. The red circles highlight the individual larvae in the wells

To ascertain if the ability to rescue larval development is unique to bacteria, we inoculated the standard larval diet with 100 μl of an active baker's yeast culture and found that this too rescued development (Table 1; line 13). These data suggest that both live bacteria and fungi are equally capable of supporting larval development. Several observations pointed to the need to develop a novel way to deliver nutrients to the mosquito in order to support axenic growth: first, high concentrations of supplements in the liquid media were often toxic to the larvae (Table 1), suggesting mosquito larvae may be sensitive to high concentrations of particular compounds in their environment. Second, based on the lack of turbidity of the larval growth medium, *E. coli* was present in relatively low numbers in the larval external environment, suggesting the cells were likely present inside the gut of the larvae. Lastly, *E. coli* was readily cultured from the larvae in densities of $10^4$ to $10^6$ cells per larvae, suggesting a high internal bacterial load. Consequently, we hypothesized that the ability of *E. coli* to rescue development was based on a high concentration of cells within the mosquito gut. Thus, we predicted that embedding heat-killed bacteria in a solid matrix might rescue mosquito development by protecting the larvae from exposure to high concentrations of bacterial components in the media and delivering any essential bacterially derived nutrients directly to the gut. Thus, the standard mosquito food consisting of the liver and yeast extract (hereafter, LY), and heat-killed bacteria were embedded in an agar plug. This diet was capable of supporting larval development to pupation (Table 1, line 14; Fig. 1). To test if larval development could be supported solely on the heat-killed *E. coli* cells, the LY base was removed. On this diet, larvae failed to develop (Table 1; line 16). When the *E. coli* was removed and the larvae were fed on agar pellets only containing the LY base, larval development was also rescued, indicating that bacterial components are not required for mosquito development

(Table 1; line 16). However, axenic larvae reared on LY alone developed more slowly than the agar plugs supplemented with attenuated *E. coli* (see data below). To test if this difference was due to the additional food in the *E. coli* supplemented agar plugs, the amount of LY was doubled, but this resulted in death of the larvae (Table 1, line 17), suggesting overfeeding larvae with agar plugs can also lead to mortality. In summary, these data indicate that live bacteria are not required to rescue larval development. Furthermore, larvae can be reared in the complete absence of any bacterially derived nutrients.

**Verification of mosquito sterility**. For every experiment, we employed a three-step verification process to ensure the sterility of axenic mosquitoes as follows: (1) in each experiment, a sterile control group of larvae was maintained. This group was processed in parallel to the other treatment groups and fed only liver: yeast extract (not embedded in agar plugs), which does not support larval development (Table 1, Fig. 1). Any larval development in this group past the L1 stage was taken as an indication of contamination and the experiment was discarded (Fig. 1). (2) In each experiment, a subset of axenic and conventionally reared larvae and adult mosquitoes and agar plugs were homogenized and tested for bacterial or fungal contamination by culturing (48 h in Luria broth (LB) and yeast extract peptone dextrose (YPD) media). Conventionally, microbiome mosquitoes tested positive for the presence of both bacteria and fungi. A positive test for bacterial/fungal growth in any axenic specimen indicated

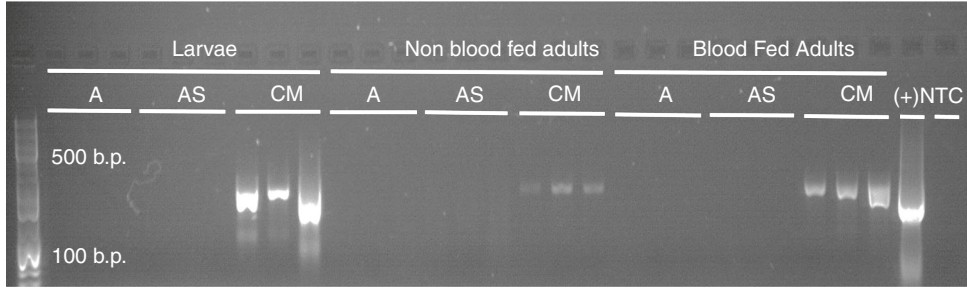

**Fig. 2** PCR detection of bacterial DNA. Total DNA was extracted from axenic (A), axenic supplemented with heat-killed *E. coli* (AS), and colony microbiome (CM) mosquitoes at three different developmental stages (larva, non-blood fed adult, and blood fed adult). DNA from three individuals from each group was used as a template for amplification of bacterial 16S rRNA genes. A and AS mosquitoes show no visible PCR products across all developmental stages, whereas amplification products were identified in the CM mosquitoes. A positive control (+) containing amplified *E. coli* K-12 DNA and a non-template control (NTC) are also included on the gel. Amplification consisted of 30 cycles and an annealing temperature of 55 °C (see Methods)

contamination and the experiment was discarded. (3) A subset of axenic larvae (L4 growth stage), newly emerged adult mosquitoes, and post blood fed adult females were tested for bacterial DNA through PCR of bacterial 16S rRNA genes. A positive test for bacterial presence in any experiment indicated contamination and the experiment was discarded (Fig. 2).

**Development of axenic and bacterial colonized larvae**. Having generated axenic mosquitoes and confirmed their sterility, we assessed if there were developmental effects associated with the axenic state by comparing axenic mosquitoes with gnotobiotic mosquitoes colonized by *E. coli* and mosquitoes exposed to the microflora of conventionally reared *Ae. aegypti* colony larvae, hereafter termed colony microbiome (CM; see Methods below). Additionally, the two axenic diets (LY only and LY supplemented with heat-killed *E. coli*) were also included to test for the developmental differences due to diet.

Out of the four conditions, the gnotobiotic group showed the lowest larval mortality with no larvae dying over the course of the experiment. In comparison, an average of 3.7% (±0.1), 5.2% (±0.1), and 27.8% (±0.2) of larvae died in the CM, supplemented axenic, and axenic groups, respectively. The axenic mosquitoes also exhibited a significant delay in time to pupation in comparison with the gnotobiotic and CM groups (Fig. 3). On average the axenic mosquitoes pupated more than 6 days after the bacterially colonized larvae. Furthermore, the difference in pupation time between the two axenic groups was significantly different (Kruskall–Wallis test, H(3) = 165.3, P < 0.0001; Fig. 3), indicating that increased nutrition may play a role in developmental time. There was no significant difference in development time between the CM (M = 5.30, SD = 0.46) and gnotobiotic groups (M = 4.87, SD = 0.40; Dunn's multiple comparisons test, P = 0.28), suggesting laboratory adapted *E. coli* is as adept at rescuing larval development as the normal flora of colony mosquitoes.

**Biometric assessment of adult mosquitoes**. To determine if our axenic rearing conditions altered mosquito phenotypic traits, we examined adult sex ratio, wing length, and survivorship. The gnotobiotic group contained the lowest proportion of females (29.4%), which was significantly different from the CM group, which displayed the highest proportion of females (54.0%; Fisher's exact test, P = 0.0157; Fig. 4a). The axenic groups were not significantly different from each other (Fisher's exact test, P = 1.000), or the bacterially colonized groups (Fisher's exact test, P > 0.05), suggesting that the axenic state had little or no effect on mosquito sex ratio (Fig. 4a). Wing length was used to assess adult size. The bacterially colonized males and females displayed larger

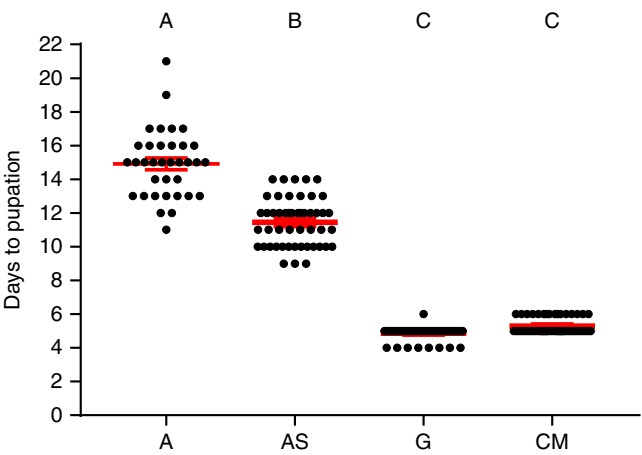

**Fig. 3** Delayed development in axenic larvae. Time to pupation is shown for axenic (A), axenic supplemented with heat-killed *E. coli* (AS), gnotobiotic (G), and colony microbiome (CM) larvae. Points represent time to pupation for individual larvae. The experiment was performed with 18 individuals per treatment (three six-well plates), with three independent replications (i.e., 54 individuals). The central red line denotes the mean pupation time, with the bars indicating the standard error of the mean. Statistically significant differences between groups were identified with the Kruskal–Wallis test and columns labeled with different letters were significantly different with a P-value < 0.05

wing lengths than their axenic cohorts (one-way ANOVA, females: F(3,70) = 84.06, P < 0.0001, males: F(3,105) = 48.34, P < 0.0001; Fig. 4b). Thus, the axenic mosquitoes appeared to be stunted in comparison with mosquitoes raised in the presence of live bacteria. Additionally, supplementing the axenic diet with heat-inactivated *E. coli* significantly increased axenic female wing size (Axenic: M = 2.75, SD = 0.07; Supplemented Axenic: M = 2.831, SD = 0.08, Tukey's HSD test, P = 0.02), but not to the same extent as mosquitoes colonized by live bacteria (Fig. 4b). Supplementing the diet did not significantly increase axenic male wing size (A: M = 2.21, SD = 0.07, AS: M = 2.24, SD = 0.01, Tukey's HSD test, P = 0.83); Fig. 4c).

Finally, survivorship of the adult mosquitoes in all four groups was assayed by investigating days of survival post pupation. Mosquitoes in the axenic group showed the highest mortality and shortest adult lifespan, surviving at most 24 days (Fig. 5a). In contrast, supplementing the axenic diet with heat-inactivated *E. coli* increased the maximum adult lifespan to >40 days (Fig. 5a). Interestingly, adult survivorship was significantly different

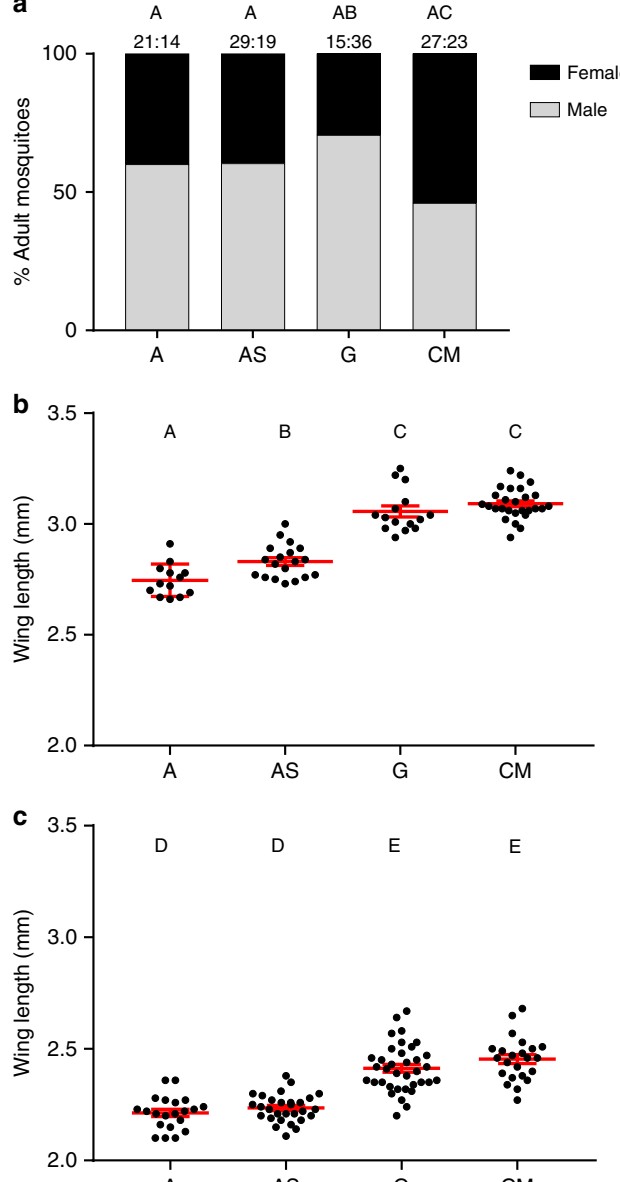

**Fig. 4** Biometric assessment of adult mosquitoes. **a** Male:female ratio for axenic (A), axenic supplemented with heat-killed *E. coli* (AS), gnotobiotic (G), and colony microbiome (CM) mosquitoes. Bars marked with a different letters were significantly different in sex ratio with a *P*-value < 0.05 by Fisher's Exact test. The number of mosquitoes assayed is indicated above the bars. **b** Female wing length. **c** Male wing length. Points represent wing lengths of individual mosquitoes (*n* = 54). Mean wing length and standard error of all individuals are signified by the red bars. The numbers of mosquitoes assayed are the same as indicated in panel A. Columns labeled with different letters were significantly different with a *P*-value of < 0.05 as determined by a one-way analysis of variance

between all groups when compared together (Logrank test *P* < 0.0001) or pairwise (Logrank Mantel–Cox). However, as discussed previously, the axenic group showed a significant delay in development in comparison with the other groups (Fig. 3). The survivorship data were adjusted to account for total lifespan by adding the larval development time to adult survivorship (Fig. 5b). In this case, the lifespan of the axenic and bacterially colonized hosts was more congruous. However, the *E. coli* supplemented axenic mosquitoes still exhibited the longest lifespan (Fig. 5b). As

above, even when incorporating the differences in developmental time the survivorship curves were significantly different between all groups. Taken together these data show that there is a complex relationship between the axenic state, diet, and longevity. Axenic mosquitoes spent more time as larvae with little change in their overall lifespan, but feeding them a diet of heat-inactivated bacteria decreased their development time (Fig. 3) and increased their lifespan (Fig. 5).

**Egg clutch size of axenic mosquitoes.** There was no significant difference in the egg clutch size between any of the groups of mosquitoes (one-way ANOVA, $F(3,50) = 0.8081$, $P = 0.50$; Fig. 6). The axenic state did not have any apparent effect on the viability of the eggs as they were readily hatched. Sterility of the subsequent generation was confirmed by culture-dependent and independent means. These data indicate that under the proper conditions mosquitoes could be reared microbial-free over multiple generations, leading to the potential establishment of axenic mosquito colonies.

**Bacterial imprinting of axenic mosquitoes.** Three bacterial isolates were cultured from colony mosquito larvae that differed in their antibiotic susceptibility so that they could be differentiated by plate counts (Table 2). The three strains were presented to either axenic larvae or axenic adults as single strains or as a mixed community composed of an equal cell number of all three strains. The phenotypes of the gnotobiotic mosquitoes varied widely. Larval exposure resulted in mortality from *ca.* 22 to 88% with average pupation times from 6 to 11 days (Table 2).

Introducing the bacteria to axenic adults is a methodological development unique to this study, as previous studies relied on pretreatment of the mosquitoes with antibiotics[37]. Bacterial strains and the mixed community were presented to axenic adults in their sugar meal (see Methods below). Every mosquito assayed was colonized by the introduced strains, suggesting that bacteria are readily taken up during adult feeding.

**Bacterial community structure in mosquitoes.** The mosquitoes colonized by the mixed community were assessed for the presence and abundance of the three stains. Every mosquito measured was found to harbor all three bacterial strains (Fig. 7). For the larvae, the strain CL.2.Tc.1 was ~10 times less abundant in the microbiome than the other two strains (Fig. 7). Similarly, strain CL.2.km.3 was less abundant than the other strains in the adults (Fig. 7). Both of these strains demonstrated the highest mortality in the larvae and adults, respectively (Table 2), suggesting there may be some modulation by the host of the composition of the microbiome. This, however, could be a measurement artifact, as we only assessed live mosquitoes. It is possible the mosquitoes that died were those where the more lethal strains gained a larger foothold in the microbiome. These data highlight the utility of manipulating the microbiome in order to characterize the effects of bacterial dynamics on the phenotype of the host.

## Discussion

Contrary to previous findings[27,35,36], we demonstrate that it is possible to rear axenic mosquitoes from larvae to adults and into the next generation. Furthermore, the colonization of axenic larvae with *E. coli*, individual strains cultured from colony mosquitoes, and a simplified microbial community all demonstrate an ability to manipulate the microbiome, specifically to imprint axenic mosquitoes with bacterial communities of a known composition. This transitions the microbiome to an experimental variable that can be utilized to gain a better mechanistic

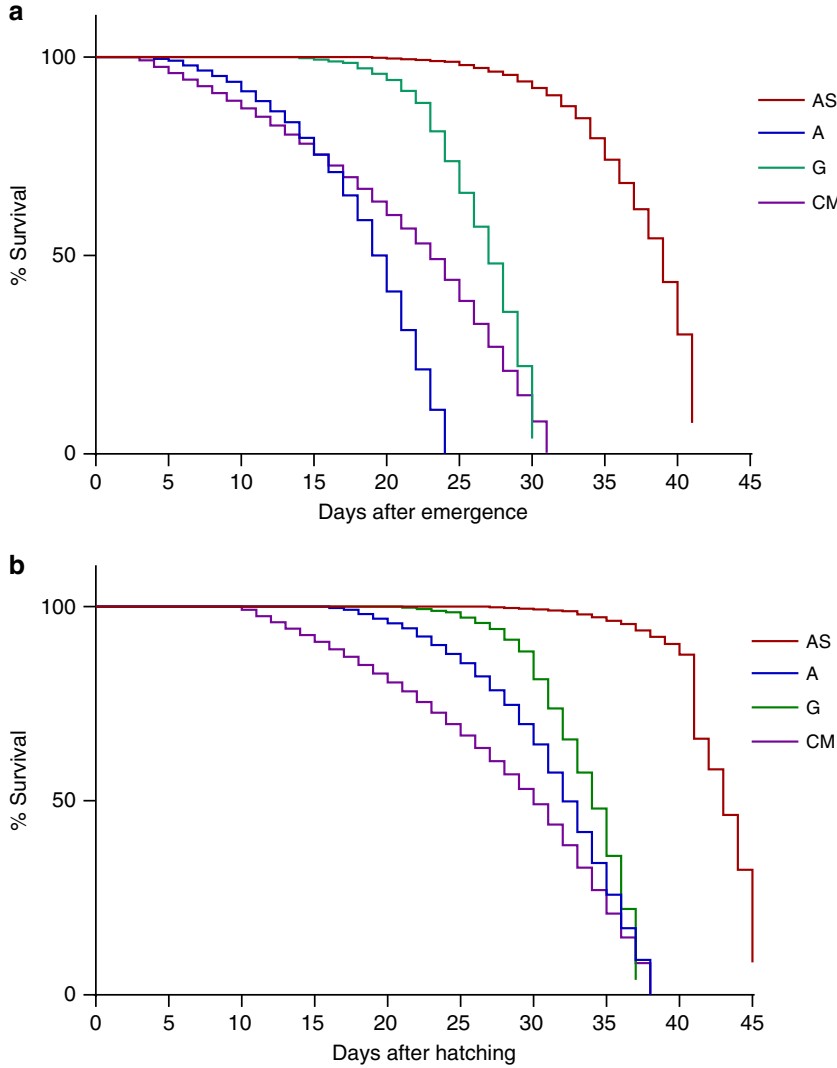

**Fig. 5** Mosquito survival curves. **a** Adult survival as measured days post emergence. **b** Mosquito survival curve corrected for larval development time. For both **a** and **b**, analysis of the Kaplan–Meier survival curves by the Logrank test revealed statistical significance when all groups were compared together ($P < 0.0001$). Sample sizes varied between groups; AS ($n = 23$), A ($n = 21$), G ($n = 17$), and CM ($n = 27$). All pairwise comparisons were also significantly different when compared using the Logrank Mantel–Cox test ($P < 0.0001$)

understanding of the interaction between the microbiome and the phenotype of the host.

Here, we show that a mixture of liver and yeast extract, when provided in high concentrations in a semi-solid form rather than freely in the mosquitoes' aquatic environment, can rescue the larval development. This suggests that the primary association between mosquitoes and their gut microbiota is nutritional rather than symbiotic, as live bacteria or fungi are nonessential to mosquito development. This raises a question as to what nutrients bacteria are providing to developing larvae in nature. A study of the transcriptional differences between axenic and colonized larvae found that axenic larvae (stalled in development due to a lack of a microbiome) displayed significant downregulation of peptidase genes and an upregulation of amino acid transporters in comparison with their microbially colonized cohorts[38]. This suggests that protein and amino acid metabolism is significantly altered in axenic larvae. A role for the microbiome in scavenging amino acids is also supported by data from axenic *Drosophila*. A fungal member of the normal microflora of the fruit fly was found to extract amino acids from nutritionally poor diets and shuttle them directly to the host[39]. Yet when we supplemented the larval diet with an amino acid and vitamin mixture, low concentrations were insufficient to rescue larval development and high concentrations were lethal (Table 1; lines 11, 12). Thus, a diet with the appropriate concentration of amino acids and proteins appears to be critical for larval development, but we have so far been unable to identify the necessary components or concentration for a fully synthetic larval diet.

Previously, bacterial-mediated hypoxia was identified as a potential mechanism to explain the apparent requirement of live bacteria for mosquito development[36]. This report was based on the observation that *E. coli* mutants in *cytochrome bd oxidase* could not rescue larval growth[36]. This oxidase has a high affinity for oxygen, and allows facultative anaerobes to maintain aerobic respiration under low-oxygen conditions[40]. In this regard, the role of this enzyme complex is generally in response to anaerobic conditions, rather than a cause of them. Thus, the inability of mutants in this complex to rescue larval development may be due to an inability of *cytochrome bd* mutants to survive anoxic conditions. Furthermore, *cytochrome bd oxidase* complexes can protect bacteria from agents synthesized by the host immune system, such as reactive oxygen species and nitric oxides[41,42].

Therefore, *cytochrome bd oxidase* mutants may be compromised in their ability to survive in host organisms[43]. Finally, our data show that mosquitoes develop in the complete absence of any detectable bacteria or fungi, indicating that microbiome-induced hypoxia is not a necessity to development.

Axenic mosquitoes demonstrated a delay in development when compared with the gnotobiotic and conventional microbiome groups. A similar delay in development has been observed in other axenic organisms, including *Caenorhabditis elegans* and *Drosophila*[44,45]. In the case of *C. elegans*, an increase in development time is usually coupled with an increase in longevity, with axenic organisms living longer than organisms reared on a non-axenic culture[44,46]. This pattern is not observed in axenic *Drosophila*, the longevity of which appears to be diet-specific[47,48]. Here, we show a complex relationship between the axenic state and diet. Axenic mosquitoes maintained on LY alone showed the lowest adult survivorship (Fig. 5a), but axenic mosquitoes fed on a diet supplemented with heat-inactivated *E. coli* exhibited a significant increase in lifespan (Fig. 5). Thus, these data suggest that the axenic state and diet interact to determine longevity in mosquitoes and the removal of the microbiome has differing

effects on host biology. This emphasizes the importance of expanding the number of axenic models to help uncover the common and diverging effects of the microbiome on host physiology.

Several of the phenotypes of the axenic mosquitoes that we observed (i.e., longer development time and stunted development) are likely due to a lack of nutrition. For example, it has been shown that *Culex molestus* mosquitoes reared on a low concentration larval diet are smaller than mosquitoes reared on a rich diet[49]. It is well established that the microbiome can act as a source of food and nutrients to the host[50,51], and bacteria are likely the primary source of food for wild mosquitoes. Only through supplementing the larval diet with high concentrations of nutrients, unlikely to be found in natural larval environments, were we able to rescue larval development. In this sense, bacteria are required for larval growth outside of a laboratory setting. However, not all roles of the microbiome are nutritional. For example, by creating a simplified two-member microbiome in *Drosophila*, researchers showed that secondary compounds produced only when the members were both present, resulted in behavioral changes for the host organism[52]. Thus, there are emergent phenotypic properties of the host that may be dependent on the composition and structure of the microbiome.

We were able to generate gnotobiotic mosquitoes differing in the composition of their microbiome ranging from a single species to a simplified microbial community (Table 2). The different gnotobiotic mosquitoes showed a wide variation in phenotypes. For example, larvae colonized by the strain CL.2.Tc.1 related to *Serratia marcescens* had >88% mortality, compared with 50% mortality in a mixed community (Table 2). An antagonistic relationship between *Serratia*-related bacteria and mosquitoes has been documented previously[53,54]. Yet, strain CL.2.Tc.1 was consistently isolated from the mosquitoes colonized by a mixed community, although in lower abundances than the other two strains (Fig. 7), suggesting that other microbiome community members may modulate the lethal effects of *Serratia*. In this regard, the composition and structure of the microbiome may be an important factor in explaining the phenotypic plasticity observed between individual mosquitoes.

In summary, this study presents a method to rear axenic *Aedes aegypti* mosquitoes from eggs to adults and into the subsequent generation in the complete absence of a microbiome. We show that axenic mosquitoes develop normally, but with a delay in the time of development. The data presented here suggest that the microbiome may be a potential target for future control strategies. Using bacteria as a tool in mosquito control, a method referred to as paratransgenesis[15,55], has already been pursued. However, these studies have thus far been hampered by a lack of effective tools to manipulate the microbiome. The methods presented in this study add mosquitoes to the collection of organisms for

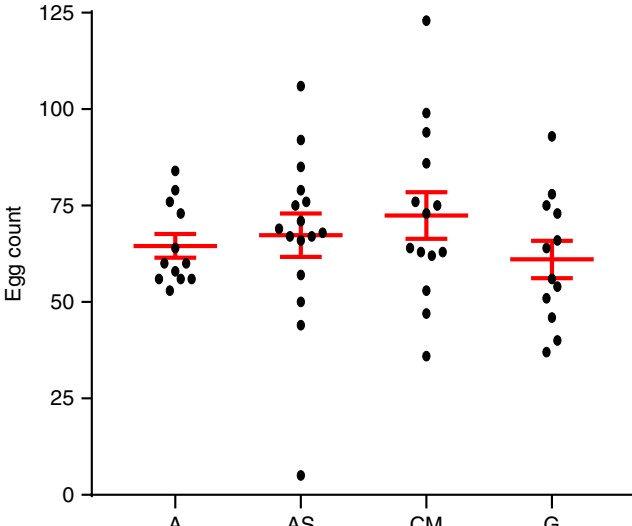

**Fig. 6** Egg clutch size. The number of eggs per mosquito is shown for axenic (A; n = 12), axenic supplemented with heat-killed *E. coli* (AS; n = 15), gnotobiotic (G; n = 12), and colony microbiome (CM; n = 14) mosquitoes. Points represent clutch size for individual females. Mean clutch size and standard error of all individuals are signified by the red bars. No significant differences between the groups was detected using a one-way analysis of variance with a *P*-value cutoff of <0.05

### Table 2 Colonization of axenic larvae and adults with simplified microbiomes

| Strain | Antibiotic resistance | Closest BLAST match (%) | Larval exposure | | Adult exposure | |
|---|---|---|---|---|---|---|
| | | | Mortality (%) | Time to pupation (days) | Colonized (%) | Mortality (%) |
| CL.3.carb.2 | Carbenicillin | *Buttiauxella izardii* strain S1-124 (99%) | 22.2 | 5.7 | 100 | 0 |
| CL.2.km.3 | Kanamycin | *Arthrobacter woluwensis* strain ED (97%) | 55.5 | 6.9 | 100 | 33.3 |
| CL.2.Tc.1 | Tetracycline | *Serratia marcescens* strain O2 (99%) | 88.8 | 11.5 | 100 | 0 |
| Mixed community | N/A | N/A | 50.0 | 7.4 | 100 | 14.3 |

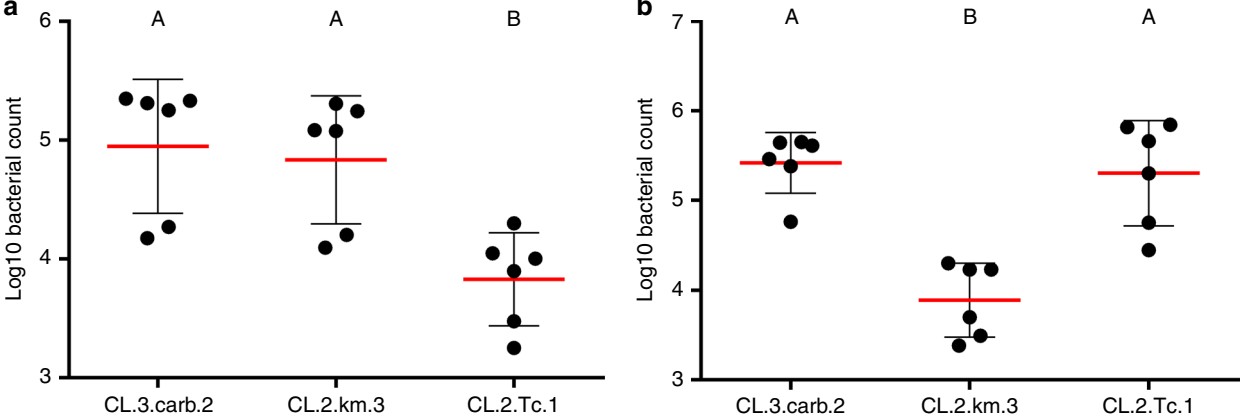

**Fig. 7** Strain abundance in mosquitoes colonized by simplified microbial communities. Each point indicates the colony forming units (cfu) of each of three strains that were introduced to individual **a** axenic larvae or **b** axenic adults. The red horizontal bar represents the mean with error bars indicating the standard error of the mean. Counts were determined from 6 individuals for both the larvae and adults. Columns labeled with different letters were significantly different as determined by a one-way analysis of variance with a $P$-value cutoff of <0.05

which an axenic state can be maintained, underpinning our ability to treat the microbiome as a controlled experimental variable in organismal studies.

## Methods

**Preparation of mosquito rearing substrates**. Multiple diets and supplements were tested for their ability to support the development of axenic mosquitoes. The first group of treatments was based on the standard diet for the colony raised mosquitoes, which consisted of a 0.1% solution of three parts liver extract (Difco, dessicated, powdered beef liver) and two parts yeast extract (Fisher Scientific, granulated yeast extract). The standard diet was also supplemented with the following: 5 ml Luria Broth (LB), 1 ml or 5 ml of an overnight culture of sonicated *E. coli* cells (throughout the manuscript *E. coli* refers to the wild-type strain K-12[56]), 1 ml or 5 ml autoclaved *E. coli* cells (overnight culture), 0.2% or 2% (w v⁻¹) yeast extract, 100 µl of an overnight culture of live baker's yeast (*Saccharomyces cerevisiae*), and a 1x and 0.5x solution of an amino acid (Gibco MEM Amino Acids 50x stock) and vitamin (Gibco Vitamin Solution 100x stock) solution mixture. Two diets included a media base not consisting of the standard diet. These were 0.1% sterile fish food (TetraminTropical Flakes) and a synthetic larval growth media[33].

**Preparation of agar plugs**. A 3.3% w v⁻¹ of a 3:2 liver:yeast extract was mixed with 1.5% w v⁻¹ agar in distilled water. The solution was autoclaved and poured into sterile Petri dishes and allowed to set. The plugs were excised from the agar plate using a sterile 15 ml conical tube equivalent to a ~0.6 g agar plug. For growth of larvae a single plug was added to each well of a six-well plate, along with 5 ml of sterile water (Fig. 1).

For the diet supplemented with heat-killed *E. coli*, two 500 ml flasks of LB broth were inoculated with *E. coli* and grown overnight at 37 °C. The *E. coli* was harvested by centrifugation at 10,000 rpm for 10 min, and the resulting cell pellet was suspended in 20 ml of PBS and autoclaved at 121 °C for 30 min. The resulting mixture was mixed with 40 ml of the 3.3% LY/agar mixture described above, the solution was poured into three Petri dishes, and plugs were generated as described above.

**Egg sterilization**. *Aedes aegypti* eggs were acquired from a colony of laboratory-reared mosquitoes (Orlando strain) maintained in environmental chambers at 28 °C with a 16:8 light:dark photoperiod. Sterilization of the *Aedes aegypti* eggs was carried out by removing a small segment of egg-covered filter paper and washing for 5 min in 70% ethanol, followed by a 5-min wash in a 3% bleach and 0.2% ROCCAL-D (Pfizer) solution, and an additional 5-min wash in 70% ethanol. The sterilized eggs were then rinsed three times in autoclaved distilled water and placed in Petri dishes filled with phosphate buffered saline (PBS). Eggs were hatched in a vacuum oven (Precision Scientific Model 29) at 25 Hz for 15 min at room temperature. A schematic diagram showing the sterilization procedure is shown in Supplementary Figure 1.

**Rearing of larvae**. For each tested condition, individual larvae were transferred from the Petri dishes to individual wells of a six-well plate. Each well of the plate contained 5 ml of the rearing substrate, or, in the case of the agar plugs, a 0.6 g plug in a 5 ml solution of sterile water. Development, time to pupation, and mortality were recorded each day for 21 days after hatching for a total of three replicate

plates (i.e., 18 individuals), which was repeated three times for biological replicates, resulting in a total of 54 individual larvae per treatment.

For the gnotobiotic group, a 10 µl aliquot of an overnight culture of *E. coli* was added to each well. For the CM group, a single larva from the laboratory-raised mosquito colony was placed in a tube containing a steel BB and distilled water and was ground using a mixer mill. A 10 µl aliquot of this mixture was added to each well (Supplementary Figure 1).

**Blood feeding**. After pupation, larvae were allowed to emerge into an autoclaved mosquito emergence chamber (see Supplementary Figure 2 for a full description). Mosquitoes were maintained on filter sterilized 10% sucrose for 4 days, after which the mosquitoes were blood fed using a circulating water bath and membrane feeder. All feeds were carried out in a biosafety cabinet under sterile conditions. Mosquitoes were fed sterile defibrinated sheep blood (Hemostat Laboratories), and axenic Swiss Webster mouse pelts (provided by Dr. Andrew Goodman, Yale University) were used as the membrane in lieu of parafilm.

**Confirming the sterility of axenic mosquitoes**. A subset of the larvae and pre and post blood fed adults were tested for the presence of live bacterial/fungal cells or bacterial genomic DNA via culturing and ribosomal gene PCR. Individual mosquitoes were transferred to a round bottom tube containing a steel BB and 150 µl of sterile PBS and homogenized for 30 sec at 30 1 s⁻¹ using a Mixer Mill. In total, 50 µl of each homogenate was inoculated into a 14-ml culture flask containing 2 ml of LB or YPD broth and incubated 48 h at 28 °C. Negative results were confirmed by an absence of bacterial/fungal growth. For the remainder of the homogenate, total DNA was extracted using a PowerSoil DNA Isolation kit (MoBio Laboratories, Inc.). DNA extractions were PCR amplified using standard 16S rRNA gene primers (27 F and 1492R[57]), using a thermocycling regime of: initial denaturation 95 °C, 10 min; denaturation 95 °C, 45 sec; annealing 55 °C, 60 s; extension 72 °C, 90 s for 30 cycles of amplification, followed by a 10 min final extension (Supplementary Figure 3).

**Biometric assessment**. To assess differences in wing length between the groups, the adult mosquitoes were anesthetized on ice and wings were removed using forceps. Wings were then visualized using a Zeiss Axioplan 2 universal microscope and wing length was measured using Axiovision (v.4.8.1) software. Egg counts were collected from mosquitoes that were previously blood fed. Mosquitoes were anesthetized using CO₂, and fully engorged females were transferred to individual 50 ml conical tubes containing a portion of sterilized coffee filter paper for egg laying. Four days post blood feed, the filter papers were removed and the eggs were counted by eye. A subset of non-engorged females from each group were transferred to sterile mosquito feeders housed in autoclaved emergence jars and assessed every day for survival for a duration of 45 days, during which time they were fed a sterile 10% sucrose solution.

**Bacterial imprinting of sterile larvae and adults**. Bacteria were cultured from homogenized colony larvae on LB media for 48 h. Colonies were isolated based on morphology and growth characteristics. Each isolate was screened for resistance to three antibiotics: tetracycline (10 µg ml⁻¹), kanamycin (50 µg mL⁻¹), and carbenicillin (100 µg mL⁻¹). Three isolates were selected such that they could be differentiated based on their growth on antibiotic plates. The isolates were identified by amplification and sequencing on their 16S rRNA gene using the universal

primers 515 F and 806R[58], and amplification products were sequenced at the Keck DNA Sequencing facility of Yale University. Accession numbers of the 16S rRNA gene sequences uploaded to NCBI are as follows: CL.3.Carb.2 (MH319112), CL.2. Km.3 (MH319113), and CL.2.Tc.1 (MH319114).

Bacteria were introduced to larvae and adults alone or in a mixed three-member community. Prior to introducing the bacteria to the mosquitoes, overnight cultures of the strains were diluted to the same optical density (O.D. 620 = 0.6) and equal volumes of each strain was mixed together to generate the simplified community. The bacteria were introduced to larvae as a 10 μl aliquot in wells of a six-well plate containing an individual larva. Each treatment consisted of 54 individual larvae and the mortality, time to pupation, and the proportion of pupae that emerged as adults was recorded. An additional group of six larvae were exposed to the mixed community and were sacrificed on day 6. The larvae were serially washed in sterile PBS to remove adhering bacteria and homogenized in sterile PBS. Serial dilutions of the homogenate were plated on antibiotic containing plates to enumerate the abundance of the three strains in the larvae.

For the adults, larvae were raised sterilely as described above, and the sterile pupae were transferred to emergence jars. After the adults had emerged, the bacteria were introduced by addition to the sugar meal. The bacteria, alone or as a mixture, were added in a 50:50 ratio to a 1% filter sterilized sucrose solution. The adults were exposed to the sucrose/bacterial mixture for 24 h after which the adults were fed on a sterile 10% sucrose solution, which was replenished every day. Five days post emergence the adults were killed, homogenized in sterile PBS, and the adults were assessed for colonization by the bacterial strains by plating the homogenate on the antibiotic plates. For the mixed culture exposure, the abundance of each strain in the adult mosquitoes was also assessed.

## Data availability

The 16S rRNA gene sequences have been uploaded to NCBI with the following accession codes: CL.3.Carb.2 (MH319112), CL.2.Km.3 (MH319113), and CL.2.Tc.1 (MH319114). The data supporting the findings of the study are available in this article and its Supplementary Information Files, or from the corresponding author upon request.

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

## Acknowledgements

We thank John Shepard and Michael Thomas for assistance with mosquito colony maintenance. We would also like to acknowledge Paul Turner of Yale University for his support of B.M.'s work in the lab, and Andrew Goodman and Natasha Barry (Yale University) for providing axenic mice for the sterile blood feeds. This publication was supported in part by the cooperative agreement Number, U01 CK000509, funded by the Centers for Disease Control and Prevention. Its content are solely the responsibility of the authors and do not necessarily represent the official views of the Centers for Disease Control and Prevention or Department of Health and Human Services. Funding was also supplied by the US Department of Agriculture Hatch Funds and Multistate Research Project (CONH00773 an NE1443) as well as the National Institute of Health, National Instistute of Allergy and Infectious Diseases (5K22AI099042-02).

## Author contributions

M.A.C. performed and designed techniques for generating and maintaining axenic mosquitoes as well as analyzing and writing up results. B.M. performed the experiments on colonizing mosquitoes with individual strains and simplified communities. D.E.B. and B.S. contributed to experimental design and writing of the paper.

## Additional information

**Competing interests:** The authors declare no competing interests.

