## [Peer Review File · Nature Communications]

Reviewers' Comments:

Reviewer #1:

Remarks to the Author:

In the present manuscript, Correa et al present a new method to derive axenic mosquitoes and use this method to assess the impact of the microbiome on mosquito development, fecundity, and growth. Previous methods to derive axenic mosquitoes have relied on antibiotics and these studies have shown that the absence of a microbiome prevents normal larval development, which has been associated with an impact of bacteria on gut oxygen levels. In this manner the present manuscript provides an important development as Correa et al have been able to rear mosquito larvae successfully to adulthood by the use of agar disks impregnated with dead bacterial cell components. In developing their axenic rearing conditions the authors observe several additional interesting features, including that high supplementation of amino acids and vitamins is lethal and that exposing axenic larvae to conventionally-reared microbes can be lethal. The reasons for this are not evident, but it provides interesting data and context to consider their results. A model for rearing axenic mosquitoes would be of great use to the community and for subsequent work identifying microbiome components that impact mosquito physiology and the mechanisms underlying these associations.

At the same time, there are a number of weaknesses in the current manuscript that should be addressed before publication.

It was often very difficult to follow the logic of the study and steps taken to arrive at their conclusions. In many cases this appears to be because the methods are not adequately explained. For example, how are the agar plugs provided and how does this method compare to typical rearing procedures or as compared to gnotobiotic or CR. Are larvae in liquid with agar plugs in the wells? In gnotobiotic and CR was the microbiome just added to wells? (What is the difference with larval media mentioned on line 99- this isn't obvious from table). There are also several statements not adequately backed by data that seem quite large jumps- line 99 assumption that bacteria are active in gut (especially when dead bacteria suffice- maybe the nutritional component just needs to be protected like in the agar plug); line 130 assumption of competition or inhibition by CR microbiome - what if this is just mimicking over-nutrition observed with amino acid or vitamin supplementation- what would the effect of autoclaved, filtered full community be on development as compared to E. coli K12 feeding?

Why not include the adult survivorship?

Why does axenic treatment include antibiotics? how then is this different than previous methods, how is this an advantage over those methods. Can the authors be sure of no antibiotic effects as these were not included in other treatments to control for the effect.

Do the authors test the agar plugs to ensure no microbial growth?

Finally, one major flaw in the current manuscript is that the authors conclusions are often contradictory. The data indicate that live bacteria are not required and their data suggest that microbes associated with mosquitoes may serve more of a nutritional role and might themselves be used for food, which the authors also conclude at some points (L228). Yet, elsewhere they argue that the bacteria are provisioning nutrients (L164). On one hand this dichotomy isn't a problem and as others have proposed (Broderick 2015 and 2016, Yamada 2015) it's likely a continuum where microbiome members can shift between roles as food and symbiont. The text would be improved by reconciling this more carefully.

Reviewer #2:

Remarks to the Author:

The manuscript by Stevens et al. attempts to provide a very relevant and important protocol for making axenic adult *Ae. aegypti*, a feat that can be performed in other insects, but to date has not been demonstrated in *Ae. aegypti*. This knowledge would be a very valuable tool for those studying how the microbiome of adult mosquitoes impacts various physiological processes, but provided the controls presented in the manuscript, I am not convinced by the protocol and require further evidence to be convinced. It is unclear to me why the axenic group of larvae is treated with carbenicillin and tetracycline but none of the other treatment groups are. This raises concerns whether the axenic treatment in fact created axenic larvae before the addition of each bacterial (dead or alive) treatment because the axenic control group may simply be bacteria free because of the presence of antibiotics. In addition, the only figure provided demonstrating the sterility of axenic group and the other treatments does not even include the treatment with the agar plugs which is the most important treatment of the manuscript. The manuscript requires the removal of the antibiotic treatment of only the axenic group and a detailed figure demonstrating the sterility of groups treated with the agar plugs.

In addition, the relevance of this protocol is sold to the reader as an important step forward in being able to imprint adult mosquitoes with whichever bacteria they are interested in to study various aspects of the role of the microbiome on mosquito physiology. I would expect a manuscript in a journal of such caliber to also perform some experiments in which they themselves do this, otherwise it simply remains a methods paper. I am not convinced the phenotypes of pupation rate of larvae and fecundity of axenic adults warrants a substantial contribution to the field since differences in development time/pupation rate and a demonstration of changes in adult phenotypes following gnotobiotic treatments have already been published (Coon et al. 2014 and Dickson et al. 2017).

We appreciate the comments of the editor and reviewers and have addressed their concerns. This has resulted in a substantially revised manuscript. We have included a marked document as reference, but we are highlighting five of the substantial changes below to help clarify the revisions. Although the revisions are substantial they have not altered the key finding of the study, namely that live bacteria are not required for mosquito development. In fact, we feel the reviewers will see that the conclusion is strengthened.

1. Both reviewers had concerns with the antibiotic treatment of the eggs. We believe that this was partly due to us not being clear that the antibiotic was only present during egg hatching but not during rearing of the mosquitoes. However, we realize this still resulted in the axenic group being exposed to antibiotics while the other groups were not (however brief the exposure was). **Thus, all the data has been repeated with no antibiotic treatment.**
2. We originally were using the agar plugs only for the axenic group but not the gnotobiotic group or conventionally reared group, where the liver:yeast extract was just added to the water. (This was to replicate “conventional rearing”). We realized this could also induce some biases in comparing the groups. All experimental data is now based on larvae fed on liver:yeast extract in agar plugs. As this is no longer based on conventional rearing the control group is now referred to as colony microbiome, as the added bacteria were derived from colony larvae (please see methods).
3. In repeating these experiments we found a concentration of liver:yeast extract that supported mosquito development without the addition of attenuated *E. coli*. **This resulted in mosquito development not only in the absence of live bacteria, but also with no bacterially derived components, further supporting that bacteria are not required for larval development.**
4. We now also have tested the axenic bacteria with bacterial 16S rRNA primers and fungal culturing, demonstrating that bacteria **and fungi** are not detected in the axenic larvae.
5. We have added an experiment documenting that bacteria can be imprinted on both the axenic larvae and adults. Further, these bacteria can range from single strains to simple communities (see Table 2 and Figure 7).

Please see below for our point-by-point response to the reviewer’s comments. Because the edits have been quite substantial we have included a marked version of the manuscript with tracked changes as well as a clean version.

Comments from the editor:

You will see from the reviewer reports that neither reviewer is convinced that the manuscript provides a protocol for generation of axenic mosquitoes. The revised manuscript would need to use a revised protocol and provide further evidence for sterility of the mosquitoes along the lines suggested by reviewers.

Response:

We believe some of the concerns were based on us not being clear on the methods. We have tried to be more concise in the method description and have provided a new analysis documenting the sterility of larvae and adult mosquitoes. (Please see specific comments below)

Furthermore, we agree with reviewer 2 that a revised manuscript would need to imprint adult mosquitoes with bacteria. Please make sure that the revised manuscript addresses these and all other concerns in full.

Response:

We have performed an experiment where we imprint the larvae with 3 individual bacterial strains and the three strains in combination, as a simplified community. The results of these experiments are presented in Table 2 and Figure 7.

Reviewer #1

It was often very difficult to follow the logic of the study and steps taken to arrive at their conclusions. In many cases this appears to be because the methods are not adequately explained. For example, how are the agar plugs provided and how does this method compare to typical rearing procedures or as compared to gnotobiotic or CR. Are larvae in liquid with agar plugs in the wells?

Response:

We have revised the methods description and now provide a photo of the experimental setup (Fig. 1). We hope this clarifies how the mosquitoes were reared.

In gnotobiotic and CR was the microbiome just added to wells? (What is the difference with larval media mentioned on line 99- this isn't obvious from table)

Response:

Yes, the gnotobiotic and CM bacteria were just added to the wells. We have substantially revised this section and presented a schematic diagram in Figure S2. We hope this clarifies how the bacteria were added.

There are also several statements not adequately backed by data that seem quite large jumps- line 99 assumption that bacteria are active in gut (especially when dead bacteria suffice- maybe the nutritional component just needs to be protected like in the agar plug); line 130 assumption of competition or inhibition by CR microbiome - what if this is just mimicking over-nutrition observed with amino acid or vitamin supplementation- what would the effect of autoclaved, filtered full community be on development as compared to E. coli K12 feeding?

Response:

The reviewer will see that the section in which line 99 previously appeared has been substantially revised. We welcome the reviewer's comments on this new section.

While we agree that this is an interesting question, there may be some difficulty in performing this experiment. As the reviewer likely appreciates, the microbiome varies between mosquitoes, thus cannot be standardized. Further, the plugs are made from a large volume of culture (1 Liter total, see methods). Thus, generating plugs from the colony microbiome would require overnight culturing of the bacteria, which would likely induce dramatic changes in the structure of the community, favoring the fastest growing cultivable strains. Thus it would be difficult to assess what exactly we were measuring. We think future studies investigating the difference between gnotobiotic strains with living bacteria and agar plugs derived from the strain would be interesting, but are best reserved for a future study.

Why not include the adult survivorship?

Response:

We have included the data on adult survivorship. (see Figure 6).

Why does axenic treatment include antibiotics? how then is this different than previous methods, how is this an advantage over those methods. Can the authors be sure of no antibiotic effects as these were not included in other treatments to control for the effect.

Response:

As both reviewers raised concerns with the antibiotic treatment there may have been some misunderstanding on what was actually done. The eggs were surface sterilized, then hatched in a petri dish containing antibiotics in a vacuum. The hatched larvae were then incubated in the presence of the antibiotics for four hours in the petri dish (the purpose to inhibit any bacteria that may have survived sterilization, which we observed in a small number of cases in initial experiments). Individual larvae were then transferred by pipetting to wells of a 6-well plate

(pipette volume ~50 ul), which we reasoned would result in very little transfer of antibiotic, which would also be diluted in the well (5 ml volume). No antibiotics were included in the 6-well plates. Thus, the larvae developed in very low antibiotic levels (see original figure 5).

However, we recognize that this procedure still results in the axenic group being exposed to antibiotics that the other groups were not. **To better control for any potential antibiotic effects we repeated the experiments with no antibiotic treatment, and all of the figures and analyses have been updated based on these experiments.** We have also altered the schematic diagram to reflect the procedural changes and to better represent how mosquitoes were sterilized and reared (see Figure S2)

Do the authors test the agar plugs to ensure no microbial growth?

Response:

Yes, the agar plugs are always tested for sterility. We have added this to the manuscript.

Finally, one major flaw in the current manuscript is that the authors conclusions are often contradictory. The data indicate that live bacteria are not required and their data suggest that microbes associated with mosquitoes may serve more of a nutritional role and might themselves be used for food, which the authors also conclude at some points (L228). Yet, elsewhere they argue that the bacteria are provisioning nutrients (L164). On one hand this dichotomy isn't a problem and as others have proposed (Broderick 2015 and 2016, Yamada 2015) it's likely a continuum where microbiome members can shift between roles as food and symbiont. The text would be improved by reconciling this more carefully.

Response: We thank the reviewer for pointing out this literature. The suggested references have been added to the manuscript. The discussion has been substantially altered based on our new findings that bacteria derived compounds are not necessary for larval development and welcome the reviewer's comments on our current discussion of the results.

Reviewer #2

It is unclear to me why the axenic group of larvae is treated with carbenicillin and tetracycline but none of the other treatment groups are. This raises concerns whether the axenic treatment in fact created axenic larvae before the addition of each bacterial (dead or alive) treatment because the axenic control group may simply be bacteria free because of the presence of antibiotics.

Response:

We have addressed the concerns with the antibiotic treatment by repeating the experiments without antibiotics; please see the response to the editor and reviewer 1 above.

The only figure provided demonstrating the sterility of axenic group and the other treatments does not even include the treatment with the agar plugs which is the most important treatment of the manuscript. The manuscript requires the removal of the antibiotic treatment of only the axenic group and a detailed figure demonstrating the sterility of groups treated with the agar plugs.

Response:

The results that were presented were from the agar plug treatment. Larvae will not develop without the plugs, so it would not be possible to obtain L4 or adult mosquitoes without the plugs.

However, we have repeated the experiment (without antibiotics, as suggested) and have added a new PCR gel (Figure 2) that shows the sterility of larvae, pre and post blood fed adult mosquitoes.

In addition, the relevance of this protocol is sold to the reader as an important step forward in being able to imprint adult mosquitoes with whichever bacteria they are interested in to study various aspects of the role of the microbiome on mosquito physiology. I would expect a manuscript in a journal of such caliber to also perform some experiments in which they themselves do this (continued below).

Response:

Based on the comments by the reviewer and editor we have added an experiment where we introduce a defined set of bacteria to axenic larvae and adult mosquitoes. (see table 2 and Figure 7).

otherwise it simply remains a methods paper. I am not convinced the phenotypes of pupation rate of larvae and fecundity of axenic adults warrants a substantial contribution to the field since differences in development time/pupation rate and a demonstration of changes in adult phenotypes following gnotobiotic treatments have already been published (Coon et al. 2014 and Dickson et al. 2017).

Response:

On this point we disagree with the reviewer. Several papers report that live bacteria are essential for mosquito development. Thus, our finding that live bacteria are not required for any stage of mosquito growth or development is a finding in itself and not just a description of a method. While we recognize that other researchers have investigated gnotobiotic mosquitoes (and these studies are referenced in our manuscript), these previous studies were using a gnotobiotic model because no system was available for rearing a bacterial-free adult mosquitoes.

Additionally, our new experiments demonstrate that a microbiome of known composition can also be introduced to both axenic larvae and adults with different phenotypic outcomes. Previously, these experiments had to be performed with antibiotic treatment of the adults, thus potential effects of antibiotic treatment could not be separated from microbiome effects. Thus, our study represents an advance in the ability to manipulate and study the composition and effects of the microbiome on mosquito biology. In this regard, the results of this study are much more than the reporting of a new method.

Reviewers' Comments:

Reviewer #1:

Remarks to the Author:

The authors have done adequate job of addressing reviewer concerns. The revision clarified the aspects that were confusing or seemed contradictory. The new experiments included in the revision, especially the treatments without antibiotics and validation of germ-free status, strengthen the authors conclusions. Relatedly, the expanded biometric data provides important support of the axénique model and that bacteria are not essential to development and survival.

REVIEWERS' COMMENTS:

Reviewer #1 (Remarks to the Author):

The authors have done adequate job of addressing reviewer concerns. The revision clarified the aspects that were confusing or seemed contradictory. The new experiments included in the revision, especially the treatments without antibiotics and validation of germ-free status, strengthen the authors' conclusions. Relatedly, the expanded biometric data provides important support of the axenic model and that bacteria are not essential to development and survival.

Response:

We thank the reviewer for their efforts in reviewing the manuscript and our revisions.